# The role of critical care nurses in shared decision-making for patients with severe heart failure: A qualitative study

**Noriko Inagaki**[1¤a]*, **Natsuko Seto**[1], **Kumsun Lee**[1], **Yoshimitsu Takahashi**[2], **Takeo Nakayama**[2], **Yuko Hayashi**[1¤b]

1 Graduate School of Nursing, Kansai Medical University, Hirakata, Osaka, Japan, 2 Department of Health Informatics, School of Public Health, Kyoto University, Kyoto, Japan

¤a Current address: Faculty of Nursing, Setsunan University, Hirakata, Osaka, Japan
¤b Current address: Osaka Medical and Pharmaceutical University, Takatsuki, Osaka, Japan
* noriko.inagaki@nrs.setsunan.ac.jp

**Data Availability Statement:** All relevant data are within the paper and its Supporting Information files.

**Funding:** This work was supported by the Japan Society for the Promotion of Science (JSPS)

## Abstract

### Aim

Patients with severe heart failure undergo highly invasive and advanced therapies with uncertain treatment outcomes. For these patients, shared decision-making is necessary. To date, the nursing perspective of the decision-making process for patients facing difficulties and how nurses can support patients in this process have not been fully elucidated. This study aimed to clarify the perceptions of critical care nurses regarding situations with patients with severe heart failure that require difficult decision-making, and their role in supporting these patients.

### Methods

Individual semi-structured interviews were conducted with 10 certified nurse specialists in critical care nursing at nine hospitals in Japan. A qualitative inductive method was used and the derived relationships among the themes were visually structured and represented.

### Results

The nurses' perceptions on patients' difficult situations in decision-making were identified as follows: painful decisions under uncertainties; tense relationships; wavering emotions during decision-making; difficulties in coping with worsening medical conditions; patients' wishes that are difficult to realize or estimate; and difficulties in transitioning from advanced medical care. Critical care nurses' roles were summarized into six themes and performed collaboratively within the nursing team. Of these, the search for meaning and value was fundamental. Two positions underpin the role of critical care nurses. The first aims to provide direct support and includes partnerships and rights advocacy. The second aims to provide a holistic perspective to enable necessary adjustments, as indicated by situation assessments and mediation. By crossing various boundaries, co-creating, and forming a good circular relationship in the search for meaning and values, the possibility of expanding treatment and recuperation options may be considered.

KAKENHI through a Grant-in-Aid for Scientific Research (C), Number JP18K10331 (NI) (https://www.jsps.go.jp/english/e-grants/index.html). The funder had no role in study design, data collection and analysis, decision to publish, or preparation of the manuscript.

**Competing interests:** We have read the journal's policy and the authors of this manuscript have the following competing interests: TN received research grants from I&H Co., Ltd., Cocokarafine Co., Ltd., and Konica Minolta Inc.; consulting fees from Otsuka Pharmaceutical; honoraria from Pfizer Japan INC., MSD K.K., Chugai Pharmaceutical Co., Takeda Pharmaceutical Co., Janssen Pharmaceutical K.K., Boehringer Ingelheim International GmbH, Eli Lilly Japan K.K., Maruho Co., Ltd., Mitsubishi Tanabe Pharma Co, Novartis Pharma K.K., Allergan Japan K.K., Maruho Co., Ltd., Novo Nordisk Pharma Ltd., TOA EIYO Ltd., Dentsu co., ONO PHARMACEUTICAL CO., LTD., GSK plc, Alexion Pharmaceuticals, Inc., and Cannon Medical Systems Co.; stock options from Bon Bon Inc.; donations from CancerScan and YUYAMA co. The other authors declare that they have no conflicts of interest to disclose. This does not alter our adherence to PLOS ONE policies on sharing data and materials

## Conclusions

Patients with severe heart failure have difficulty participating in shared decision-making. Critical care nurses should collaborate within the nursing team to improve interprofessional shared decision-making by providing decisional support to patients that focuses on values and meaning.

## Introduction

Patients with severe heart failure often require highly invasive and advanced therapies, including life-sustaining treatments. The treatment outcomes are characterized by high degrees of uncertainty. Furthermore, advances in medical technology have prolonged life but also the duration of the disease and increased the number of treatment options, thereby necessitating treatment decisions based on both the medical viewpoint and the patient's perspective via a shared decision-making (SDM) process [1, 2]. At present, several types of mechanical circulatory support (MCS) are increasingly being used to temporarily assist circulation in cases of sudden deterioration-induced cardiogenic shock. However, high-quality evidence for their usage is limited [3]. Although MCS may be temporarily life-saving, ethical issues pertaining to the duration of its continuity for sustaining life remain [4, 5]. Conversely, for patients with advanced heart failure with a gradually worsening course or for those who survived a cardiogenic shock, a left ventricular assist device (LVAD) implantation may be an option as a bridge to heart transplantation (HTx) or as a destination therapy. However, these patients are more likely to experience adverse complications following the placement of an implantable LVAD, which can substantially and adversely affect their medical condition and health-related quality of life, leading to a greater burden of care [6, 7]. Japan has a serious shortage of donor hearts. Insurance coverage for implantable LVAD as a destination therapy (and not intended as a bridge to transplantation) was finally approved in 2021 [8].

Internationally, the challenges of heart failure are being increasingly recognized. In Japan, a rapidly aging population has resulted in an increased number of older adult patients requiring treatment for heart failure. Considering these, system innovations in healthcare delivery are warranted [9]. SDM has become increasingly complex, especially in patients with severe heart failure. Hence, optimal methods need to be clarified [10].

SDM is a model that emphasizes patient participation and was initially developed for the patient-physician relationship [11]. Nevertheless, a broader concept of SDM has been proposed given the involvement of multiple individuals in the patient's decision-making process in complex real-world situations [12]. Moreover, interprofessional SDM (IP-SDM) models have been developed to emphasize the involvement of healthcare teams [13, 14], including a team approach for SDM in the critical care domain [14, 15].

To date, the nurses' unique roles within the IP-SDM process have not been well defined. Previous studies that examined the nurses' roles in SDM positioned nurses as intermediaries between physicians and patients [16–18]. Inagaki reported that although nurses were not fully involved in SDM, independent nursing support facilitated IP-SDM efforts, and a structured approach led to SDM with patients, families, and healthcare professionals [19]. Understanding the unique roles that nurses play in decision-making support may lead to the development of different approaches for managing complex SDM for difficult decisions and critical timing [20].

To develop the unique contributions and roles of nurses in SDM beyond those of mediators, it is necessary to clarify the nurses' perceptions of patient difficulties, especially for those

patients who require complex treatment decisions, and to determine the nursing perspective related to the provision of decision-making support. The American Association of Critical Care Nurses' synergy model for patient care, which focuses on the patient-nurse relationship, was developed to reexamine the unique role of nurses in the development of multidisciplinary healthcare systems [21]. This synergy model aims to optimize outcomes by linking individual patient characteristics to specific nurse competencies [21]. The patient-nurse relationship in this synergy model could be used as a reference for developing a unique nursing role to support patients in SDM.

Therefore, this study aimed to determine the perceptions of critical care nurses regarding the difficult situations faced by patients with severe heart failure and to clarify the nursing perspective in providing SDM support to patients.

## Methods

### Design

This study used a descriptive exploratory qualitative design and conformed to the Standards for Reporting Qualitative Research guidelines [22].

### Operative definitions

This study used several operative definitions as follows:

- Difficult patient situations in decision-making: This refers to the difficult situations associated with treatment decisions that patients with severe heart failure face in a critical care setting, including contextual situations and situations involving family members, healthcare professionals, and other persons or groups who may influence treatment decisions [23, 24].

- Nurse's role: This refers to the intentional engagement of the nurse as an individual and as a member of the nursing team in the SDM process for optimal treatment decisions, with the aim of supporting patients and their families in decision-making.

- SDM: This refers to a process for making choices wherein clinicians involved in the provision of treatment share information and discuss options to enable patients and their families to make treatment choices with the aim of achieving a consensus. This may include decisions regarding treatment, implementation, and subsequent evaluation [12, 13, 25].

### Participants and setting

Inagaki conducted semi-structured interviews with 10 certified nurse specialists in critical care nursing (CCNS) at nine hospitals in Japan to clarify the extent to which critical care nurses participate in the SDM process of patients with severe heart failure [19]. The CCNS credential in Japan is an advanced practice nurse certification and requires a master's degree in critical care nursing [26]. As in other countries, certified nurse specialists in Japan provide high-level nursing care to individuals, families, and populations with complex and difficult-to-resolve nursing problems. Therefore, CCNS were deemed the most appropriate target population for investigating the current situation related to difficult decision-making and nursing support.

The present study was a reanalysis of Inagaki's interviews [19]. Participants were selected via snowball sampling. The inclusion criteria were as follows: 1) CCNS who had completed the five-year renewal of their critical care nursing certification with the Japanese Nursing Association; 2) CCNS with experience in supporting patients with severe heart failure during decision-making; and 3) CCNS working at a hospital at the time of study participation. The first

**Table 1. Participant characteristics.**

| ID | Sex | Age (years) | Years of experience in CCNS | Years of experience in nursing | Types of clinical work mentioned in the interview | Implantable LVAD certified facilities | HTx certified facilities | Interview time |
|---|---|---|---|---|---|---|---|---|
| A | Woman | 51 | 5 | 29 | Emergency department | | | 51 min |
| B | Man | 52 | 11 | 29 | Cardiovascular surgical ward | | | 40 min |
| C | Woman | 42 | 12 | 19 | Cardiac care unit and Cross-departmental activities | | | 71 min |
| D | Woman | 44 | 8 | 22 | Palliative care team member and Cross-departmental activities | | | 65 min |
| E | Woman | 52 | 10 | 30 | Cross-departmental activities | | | 66 min |
| F | Woman | 51 | 10 | 29 | Cross-departmental activities | | | 76 min |
| G | Woman | 52 | 10 | 31 | Cross-departmental activities | | | 68 min |
| H | Man | 40 | 5 | 15 | Intensive care unit | ✓ | | 57 min |
| I | Woman | 50 | 9 | 28 | Palliative care team member | ✓ | ✓ | 57 min |
| J | Man | 40 | 9 | 17 | VAD team member | ✓ | ✓ | 40 min |

CCNS, certified nurse specialist(s) in critical care nursing; LVAD, left ventricular assist device; HTx, heart transplantation; VAD, ventricular assist device.

participant in the snowball sampling was a woman with access to a wide network of CCNS involved in decision support for patients with severe heart failure. After recruiting 10 participants, the sampling was terminated following data saturation. Data were considered saturated once the participants provided insights into decision-making support for 14 patients with severe heart failure, reflecting their respective institutional backgrounds. Hence, the same themes were obtained repeatedly.

Participant characteristics are summarized in Table 1. Participants were CCNS who worked in the cardiovascular surgical and cardiology wards, intensive care units, and emergency departments. Their job descriptions included single-ward affiliations, cross-disciplinary activities, and medical team activities. Additionally, CCNS affiliated with facilities accredited to provide implantable LVAD and HTx were included.

Key patient characteristics that were mentioned in the interviews, including the patient's age, sex, key family members, main treatment, and decision-making issues, are summarized in Table 2. Patients, aged ≥65 years, tended to have had several instances of acute exacerbations of chronic heart failure and had considered different treatment options (such as transfer to a different hospital or to in-home care while continuing intravenous inotropic drugs). Patients aged <65 years were those that required MCS and may progress to needing an implantable LVAD or HTx. Treatment options mentioned during the interviews included inotropic drugs, opioids, and mechanical ventilation as part of palliative care for older patients and MCS for patients aged <65 years. The decision-making issues disclosed were complex and varied, including those involving healthcare professionals and family members.

## Data collection

Data were collected face-to-face through a semi-structured interview with each participant from July 2019 to September 2019. The interviews were conducted by the lead author (N.I.), a doctoral student experienced in conducting semi-structured interviews. The average duration of the interviews was 59 min (range: 40–76 min). The interviews were recorded on a digital voice recorder with the permission of the participants and transcribed verbatim. The critical incident method-based interview guide was used for the interviews [27, 28]. This facilitated recall of extreme cases of significant events or behaviors, such as situations that went

**Table 2. An overview of patient decision-making, as described in interviews.**

| Patient attributes | | Key family members | Setting | Clinical course | Major treatments | | | | Treatment choice | Issues in decision-making | CCNS ID† |
|---|---|---|---|---|---|---|---|---|---|---|---|
| Age (years) | Sex | | | | Opioid | MV | Circulatory support | Others | | | |
| 65–79 | Man | Spouse / Child | Ward | Acute exacerbation of CHF | | | Inotropes | | Transition to home care with continuous inotropes | The attending physician does not consider transitioning to home, only transferring to a hospital | E-1 |
| 65–79 | Woman | Child | CCU | Acute exacerbation of CHF | | | Inotropes | | Transition to home care with continuous inotropes | Patient wants to transition home; however, daughter lacks confidence to support home care | I-1 |
| >80 | Man | Spouse | Ward | Acute exacerbation of CHF | ✔ | | Inotropes | | Opioid administration for dyspnea | Patient and spouse are exhausted from repeated hospitalization | B-1 |
| 65–79 | Woman | Spouse / Child | CCU | Acute exacerbation of CHF | ✔ | | Inotropes | | Inform patient of medical condition to discuss transfer to hospice | Unable to discuss palliative care because the family members do not wish to inform the patient | G-1 |
| 65–79 | Man | Spouse | Ward | Acute exacerbation of CHF | ✔ | NPPV | Inotropes | | Inform patient of medical condition to discuss palliative care | Patient is angry at not being told the truth because the spouse does not wish to inform the patient | D-1 |
| 65–79 | Man | Spouse | CCU | Acute exacerbation of CHF | ✔ | NPPV | Inotropes | CRRT | How far to treat the patient when the condition worsens | Patient wants aggressive treatment; however, medical professionals have questions | C-1 |
| >80 | Man | Children | ED | Acute exacerbation of CHF | | NPPV | | | Intubation for dyspnea | Patient does not want to be intubated; however, his sons want to intubate him | F-1 |
| 65–79 | Man | Spouse / Child | ICU | Acute exacerbation of CHF | | IPPV | IABP | | Transition from invasive treatment to palliative care | Patient does not want invasive treatment; however, cannot tell family about it | H-1 |
| >80 | Woman | Children | Ward | Deterioration due to severe AS | | IPPV | | AVR | Open heart surgery (AVR) for AS | After the patient's rapid deterioration, the surgical option became ambiguous | F-2 |
| 65–79 | Man | Spouse / Child | ICU | Cardiogenic shock due to ACS | | IPPV | IABP VA-ECMO | PMR repair CRRT | Resume mechanical ventilation due to septic shock | Everyone agreed not to resume mechanical ventilation; however, the patient's deteriorating condition swayed the decision | E-2 |
| 35–64 | Man | Cousin / Parent | ICU | Cardiogenic shock due to ACS | | IPPV | VA-ECMO Open chest ECMO Paracorporeal VAD | DC | Proceed to paracorporeal VAD treatment | Decision on paracorporeal VAD imminent without confirmation of patient's intent | H-2 |
| 35–64 | Woman | Spouse | ED | Acute onset CPA | | IPPV | VA-ECMO | | ECMO withdrawal | Spouse requested withdrawal of ECMO; however, medical consensus is needed as the condition is terminal | A-1 |

*(Continued)*

**Table 2.** (Continued)

| Patient attributes | | Key family members | Setting | Clinical course | Major treatments | | | | Treatment choice | Issues in decision-making | CCNS ID[†] |
|---|---|---|---|---|---|---|---|---|---|---|---|
| Age (years) | Sex | | | | Opioid | MV | Circulatory support | Others | | | |
| 18–34 | Woman | Spouse | CCU | Cardiogenic shock due to fulminant myocarditis | | IPPV | pVAD<br>VA-ECMO | | Transfer to a more advanced medical care facility | If the patient's condition does not improve, long-term support by the family needs to be considered, with a view of HTx | C-2 |
| 35–64 | Man | Parent Sibling | Ward | Gradually worsening<br>CHF | | | Inotropes | | Implant LVAD and proceed to HTx | Patient transferred to HTx facility without family consensus on HTx | I-2 |

AS, aortic stenosis; ACS, acute coronary syndrome; AVR, aortic valve replacement; CCNS, certified nurse specialist(s) in critical care nursing; CCU, cardiac care unit; CHF, chronic heart failure; CPA, cardiopulmonary arrest; CRRT, continuous renal replacement therapy; DC, decompressive craniectomy; ED, emergency department; HTx, heart transplantation; IABP, intra-aortic balloon pump; ICU, intensive care unit; IPPV, invasive positive pressure ventilation; LVAD, left ventricular assist device; MV, mechanical ventilation; NPPV, noninvasive positive pressure ventilation; PMR, papillary muscle rupture; pVAD, percutaneous ventricular assist device; VAD, ventricular assist device; VA-ECMO, veno-arterial extracorporeal membrane oxygenation.

[†] The footnote symbol before the hyphen indicates the CCNS ID, and the number after the hyphen indicates the order of the cases mentioned in detail in the interview. Cases that were not discussed in detail were excluded.

particularly well or that were particularly difficult. Participants were able to provide detailed descriptions and focus on the following open-ended questions: "How involved were you in the cases in which you felt you were fully involved in the decision-making process of patients with severe heart failure?", "Conversely, in cases where it was difficult to be fully involved, what were the difficulties?", and "Why do you continue to be involved in difficult decision-making?"

The lead author has experience as a CCNS in providing decision support to patients with severe heart failure, which may have influenced data collection and interpretation. Therefore, the experiences of the lead author were set aside to allow the participants to freely narrate their own experiences and thoughts, which were subsequently interpreted with care as their views.

## Data analysis

For data analysis, we combined reflexive thematic analysis by Braun and Clark [29] with affinity diagramming [30]. Reflexive thematic analysis comprises six phases: 1) familiarizing oneself with the data, 2) generating initial codes, 3) searching for themes, 4) reviewing themes, 5) defining and naming themes, and 6) producing the report. In this study, the affinity diagramming method was used from 2) generating initial codes to 5) defining and naming the themes. Affinity diagramming is used internationally in various fields, including healthcare [31–33]. Affinity diagramming was derived from the KJ method, which was developed by the Japanese ethnologist, Kawakita [30, 34]. In this study, we anticipated that the participants would discuss various cases and contexts; hence, affinity diagramming was considered as it aims to integrate the fragmentary information of several cases to provide overarching themes and concepts that offer new insights. The lead author attended multiple training sessions on this method, conducted the survey and analysis, and discussed the entire process with two other authors (Y.H. and N.S.) with experience teaching this analytical method.

Affinity diagramming involves three essential processes: 1) label making, 2) label grouping, and 3) chart making [30, 35, 36]. The flow of the analysis in this study is illustrated in Fig 1.

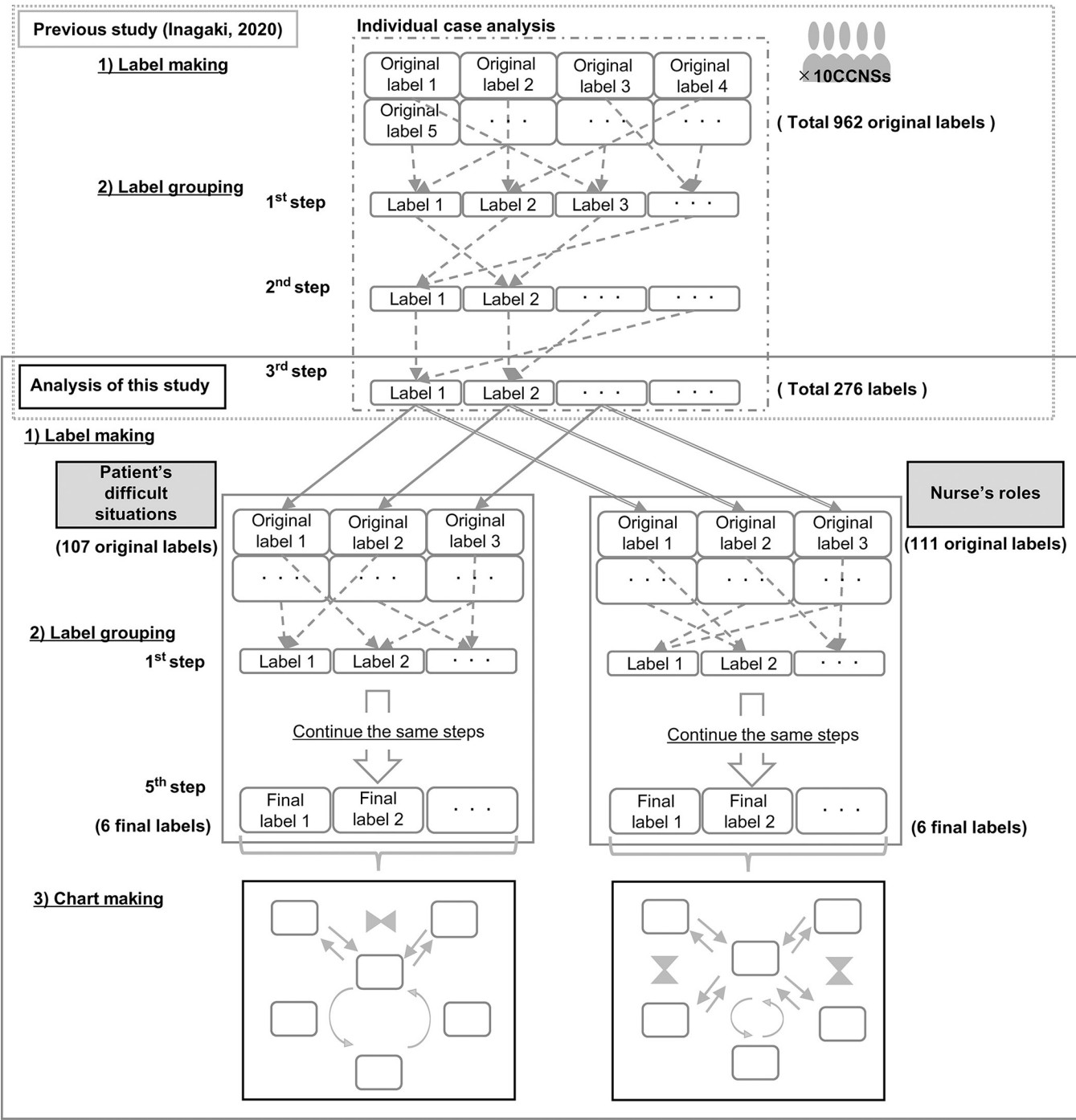

**Fig 1. Flow of analysis by affinity diagramming.** Note. 1) Label making: The qualitative data are divided into units that contain a single semantic content, and an original label is created for each unit. The original labels of Inagaki's previous study [19] revealed "the actual situation of nurse participation in shared decision-making for patients with severe heart failure." In this study, original labels were created for each patient's difficult situations and nurses' roles. 2) Label grouping: In the label grouping process, all cards are shuffled and arranged; the cards are repeatedly read to grasp the meaning and grouped according to affinity or similarity rather than preconceived notions into labels that denote the common theme of the group in a single sentence. Subsequently, the labels are further grouped. This step of labeling the top groups is repeated until the final group comprises five to seven labels. 3) Chart making: The logical relationships among the final labels are explored until a consensus is reached among the researchers and is diagrammatically presented.

First, in label making, a total of 962 original labels were created on "the actual situations in which nurses participated in the SDM of patients with severe heart failure" based on interviews with the 10 CCNS in Inagaki's study [19]. Second, for label grouping, all created cards were shuffled and arranged; the cards were repeatedly read to grasp their meanings and grouped according to affinity or similarity rather than preconceived notions. Subsequently, from the 276 labels in the third step grouping of the previous study, 107 original labels that related to "patient's difficult situation" and 111 original labels designated as "nurse's role," as perceived by the CCNS, were created. Third, in chart making, the logical relationship of the six labels was put into a diagram, and themes that symbolized the condensed content of each group were named with consensus among all the researchers.

## Trustworthiness

The rigor and trustworthiness of this study were established using Lincoln and Guba's criteria of credibility, confirmability, dependability, and transferability [37]. To ensure credibility, the details of 14 different cases from 10 CCNS with various job roles at nine different hospitals were collected to capture a complete and triangulated picture from multiple perspectives. To obtain critical feedback on the results, the themes, final labels, and lower labels that emerged from the collected data were member-checked by nine participants, and the results were revised by the researchers. The revised results were peer-debriefed by two university faculty members who specialized in critical care nursing. Then, the results were further revised. Member-checks and peer debriefings were conducted between October 2021 and January 2022. Confirmability and dependability were established through repeated discussions of the analyzed data among the researchers with different backgrounds and expertise. Transferability was established by providing a detailed and in-depth description of the background of the participants and the narrated cases, and the setting and context in which the phenomenon occurred.

## Ethical considerations

This study was approved by the Ethics Review Committee of Kansai Medical University (approval number: 2021187). Written informed consent had already been obtained from all participants during Inagaki's interview study [19]. Participants were notified via e-mail about the reanalysis using the previous survey results and could opt out. Moreover, to increase the credibility of the study results, participants were informed that the data would be provided to two external experts for review after processing to ensure that they could not be identified. None of the participants opted out, and all data were included in the reanalysis.

## Results

### Patients' difficult situations in decision-making

The perceptions of the interviewed nurses regarding the difficult situations in treatment decisions of patients with severe heart failure in the critical care setting were organized into six themes, and their interrelationships are presented in Fig 2. First, treatment decisions that may affect the patient's future life were often made without patient involvement. Yet, the patient is central to his or her own precious life. Patients were not included because of 1) "painful decisions under uncertainties" and 2) "tense relationships among the patient, family, and health-care professionals." In addition, 3) "wavering emotions during decision-making," 4) "difficulties in coping with worsening medical conditions," and 5) "patient's wishes that are difficult to realize or estimate" created a vicious cycle that easily "closed the minds" of the patients and their families. Furthermore, the need for life support-related equipment

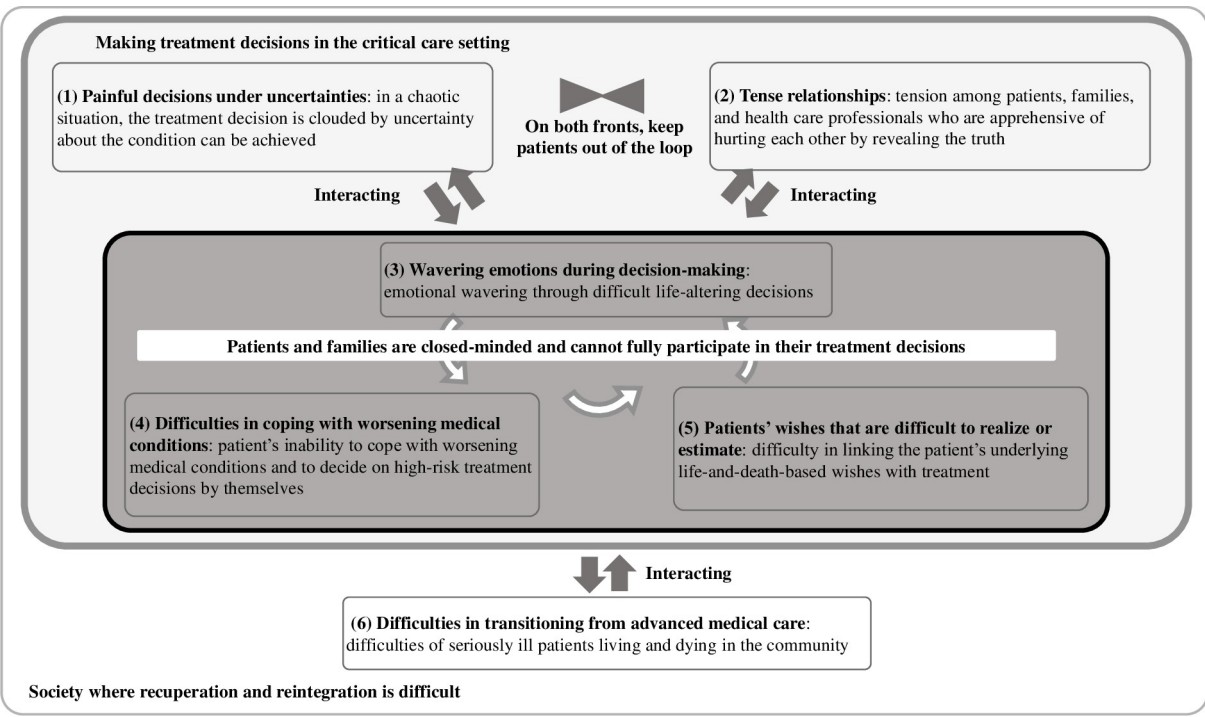

**Fig 2. Difficult situations in decision-making for patients with severe heart failure.** Note. This chart illustrates the six themes related to difficult situations in decision-making for patients with severe heart failure and the relationships at play in these situations. Unidirectional arrows between the labels indicate the order in which things occurred and/or the direction of influence. The vertices of the two triangles facing each other indicate a pairwise relationship between the content on both sides.

management skills may lead to 6) difficulties in transitioning from advanced medical care. In the next section, we describe the themes in more detail and report excerpts from the participants' interviews (see also S1 Table).

**1) Painful decisions under uncertainties.**  Treatment decisions in critical care settings involve painful decisions that may be associated with uncertainties. In an imminent life-or-death situation, patients with severe heart failure and their families are forced to make treatment decisions without knowing whether the treatment can save their lives or the conditions that they may expect. In some cases, the patient's consciousness could not be confirmed.

*If initiated as a bridge to recovery or transplantation, extracorporeal VAD may still be a better choice, even if there are uncertainties. Nevertheless, in the worst-case scenario, the patient may not be weaned off, the indication for transplantation may not be approved, and the patient may not be discharged with an extracorporeal VAD or may not survive. (CCNS: H)*

**2) Tense relationships.**  In critical care situations, when treatment decisions need to be made, tension can arise between people because they fear that the truth may hurt the patient. The severity of the illness may also create tension among patients, families, and healthcare professionals, making it difficult to openly discuss the diagnosis and treatment options, which, in turn, may affect life or death.

*Are you asking me to tell a patient that you are going to die? Why should I tell him to lose hope in such a situation? The doctor did not want to discuss the matter at all, and I thought it was impossible to have a calm discussion with them. (CCNS: F)*

**3) Wavering emotions during decision-making.**   The emotions of patients with severe heart failure or of their families can fluctuate widely during difficult, life-threatening decision-making. When critically ill patients make a treatment decision or when their family members decide on their behalf, there is usually no "grace period"; even if the patients and their families are not fully prepared. Moreover, following treatment, if the patient's condition deteriorates owing to complications, the patient and/or their family may feel regret or guilt for having made the decision, which adds to the emotional turmoil throughout the decision-making process.

*Families who witness patients suffering before a LVAD implantation surgery think that they can do something about it, but many families suffer from the inability to support the patients after surgery. Most frequently, family members are unable to support the patients who suddenly deteriorate and there is no time to talk with other family members. (CCNS: J)*

**4) Difficulties in coping with worsening medical conditions.**   High-risk treatment decisions must be made, however some patients have difficulty coping with their deteriorating medical status during the decision-making process. Patients do not cope well with accepting that their symptoms may not improve. Patients also find it difficult to accept their unstable medical condition and the occurrence of treatment-associated complications. Furthermore, they have difficulties discussing their treatment and recuperation with their healthcare professionals.

*The patient told the doctor, "If there is any treatment that can be done, I want it all done." The same patient told the nurses, "I don't want anything else to be done," and "I do not want to live any longer." (CCNS: C)*

**5) Patients' wishes that are difficult to realize or estimate.**   In some cases, the patient's wishes may conflict with treatment. It is also difficult to realize a patient's strong will, which may be the result of their struggle with the disease and their views of life and death. Family members may also have difficulty predicting the wishes of nearly unconscious patients.

*The family doctor decided that "if they persisted any longer, they would have to be sent to an emergency room" and forcibly obtained consent from the patient to be admitted to the hospital. Reflecting on the process up to that point, it seemed to me that the patient wanted to stay at home if at all possible. (CCNS: E)*

**6) Difficulties in transitioning from advanced medical care.**   Critically ill patients often have difficulties living and/or dying in the community while continuing to receive advanced medical care. For example, severely ill patients find it difficult to receive community care. These difficulties may be associated with gaining support for understanding and living with advanced heart failure (the stage and the condition) and transitioning to in-home care with continual treatment, including life support-related equipment management.

*Finally, we found a family doctor who was willing to manage inotropes at home. I talked to the attending physician, "This is the only chance for the patient can go home. Why don't we go ahead and talk about this option, even if the patient will eventually need to be readmitted?"*

*However, the attending physician thought that inotrope management at home was dangerous, and, in the end, it did not work out. (CCNS: G)*

### The roles of critical care nurses in shared decision-making

The roles of critical care nurses in decision support, as perceived by the interviewed CCNS, were summarized into six themes, leading to the overall picture presented in Fig 3. All identified roles were performed in collaboration within the nursing team. The roles included (1) the search for meaning and value (the foundational role). There were two main positions within these roles. The first aims to provide direct support including (2) partnership and (3) rights advocacy. The second aims to provide a holistic perspective for making necessary adjustments, as indicated by (4) situation assessment and (5) mediation. These roles complemented each other according to the difficulties and needs of the patients and were fed back into (1) the search for meaning and values. Furthermore, by crossing various boundaries and by(6) co-creating and forming a good circular relationship based on a search for meaning and values, the possibility of expanding treatment and recuperation options may arise. In the next section, we describe the themes in more detail and report excerpts from the participants' interviews (see also S2 Table).

**1) The search for meaning and values.** The central role of the critical care nurses is to support the search for meaning and value within a given situation. The patients/families are supported in their search for meaning and values within the situations brought about by the illness and treatment, in the treatment decision-making process, and in sharing the discovered meaning and values with all concerned individuals.

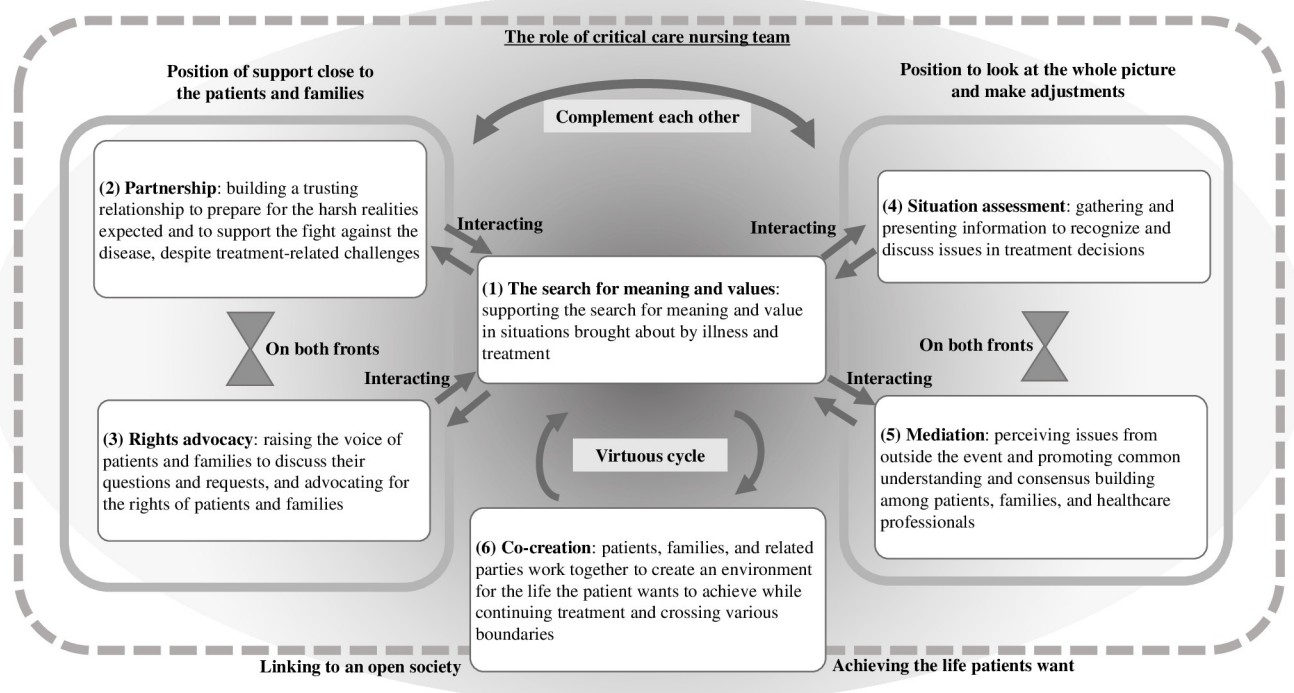

**Fig 3. The roles of critical care nurses in shared decision-making.** Note. This chart illustrates the six themes associated with the critical care nurses' roles in shared decision-making and the relationships among these situations. Unidirectional arrows between the labels indicate the order in which things occurred and the direction of their influence. Bidirectional arrows indicate the complementary relationships between them. The vertices of the two triangles facing each other indicate a paired relationship between the contents on both sides.

*During life-prolonging treatment, no one knows how much time the patient may have left to live. I sensed, from a comment made by the patient's wife during a visit, that this time meant a great deal, and I intentionally kept it in the electronic medical records to share with all healthcare professionals involved. (CCNS: E)*

**2) Partnership.** The nurses developed partnerships with the patients/families and supported them in their fight against the disease. Establishing a trusting relationship with the patient/family was deemed important, especially if the treatment was demanding. In their role as critical care nurses, the nurses aimed to support the process of choosing a treatment and providing ongoing support by sharing the emotions of the patients/families that were associated with the treatment outcomes, regardless of the treatment process-related challenges.

*I am the one who was hated preoperatively, basically for telling the facts (length of waiting time for transplantation and fatal complications after LVAD placement). But if I don't tell them the facts, I can't build a trusting relationship with them. If a fatal complication happens after the surgery, and the patient's family will be most disappointed and saddened, saying, "I didn't know this was going to happen." (CCNS: J)*

**3) Rights advocacy.** Nurses supported patients and their families by discussing their questions and needs. By providing an environment conducive to calm discussions, sometimes with the help of a neutral party, patients and their families could voice their questions and requests and defend their rights by discussing their views with others of differing opinions.

*I told the attending physician that the patient was already aware of his condition. Therefore, the patient should be informed of his condition. We discussed whether there was a way to tell the patient other than just saying, "You are going to die." (CCNS: D)*

**4) Situation assessment.** Nurses were responsible for considering the whole picture, recognizing issues in treatment decisions, and preparing for discussions. Having identified issues in treatment selection, the role of the nursing team was to gather information in a timely and focused manner, integrate and organize it, and then present it to the medical team for consideration.

*From a nursing standpoint, we can now systematically gather information for multidisciplinary conferences and future treatment decisions, such as the patient's values and/or the wishes of patients and their families. After all, we do not want to disregard the important information that would be beneficial to share at conferences. (CCNS: H)*

**5) Mediation.** As mediators, nurses perceived the problems from outside the event and facilitated a common understanding and consensus building. Having obtained an external perspective of the conflict or tension, the mediator aims to promote common understanding and consensus-building by arbitrating with the patient, family members, and healthcare professionals, provide explanations that correspond to the level of understanding and acceptance, and develop a plan for everyone to discuss the problem.

*When healthcare professionals recommend HTx to families who are unsure about it for the sake of the patient, if they do not consider the wishes of the family, then the relationship between the family and the patient may deteriorate. Everyone must discuss what to do while considering the risks and benefits of all the options. (CCNS: I)*

**6) Co-creation.** To achieve the patient's desired life, nurses participate in creating relationships with the people involved and in crossing various boundaries. For example, it is necessary to clarify the recuperation lifestyle that the patient wants and make it happen together with the patient, family members, and other concerned parties. While continuing further treatment, feasible methods are concurrently examined and nurses facilitate their realization.

*When considering the transition to home, various family concerns have arisen. Therefore, we asked the family, family doctor, visiting nurse, care manager, and all the related parties to come to the ICU together. We were able to think together about how to proceed, so we were able to move forward without misunderstanding, and, finally, the patient was able to be discharged. (CCNS: I)*

## Discussion

This study focused on patients with severe heart failure and the patient-nurse relationship during treatment decision-making. The data was derived from CCNS interviews on the patients' difficulties and the roles of nurses. The interviewed CCNS captured and expressed the difficulties faced by patients with severe heart failure in their treatment decisions, including their context. Furthermore, this study revealed that depending on the patient's difficulties, critical care nurses have various collaborative roles within the nursing team, both in working closely with the patients and their families and as overall care coordinators. Although SDM for treatment decisions has focused on the patient–physician relationship, the findings of this study revealed the importance of the unique and diverse roles of critical care nurses in promoting SDM.

The difficult decision-making situations of patients with severe heart failure identified in this study are similar to those reported in previous studies [38–42]. Similar to our results, challenges in MCS decision-making in patients with severe heart failure, including proxy decision-making, being forced to choose to live, and extreme time pressure, have been reported previously [38, 39]. Furthermore, even when patients with severe heart failure have the capacity to communicate their thoughts and opinions, they still experience difficulties coping with their illness because of depression and anxiety [40, 41]. These can render participation in the decision-making process challenging, as was evident in our study.

Patients with severe heart failure who require MCS may regain consciousness after circulatory stabilization and may not be under deep sedation. Therefore, the process of communicating life-or-death treatment to these patients and collaboratively developing a treatment plan needs further exploration. Moreover, ways and means to address the tensions and problems that arise among patients, their families, and healthcare professionals over informing patients with severe heart failure of their condition to promote patient participation in the decision-making process in the face of many challenges, need further research. Furthermore, difficulties in transitioning between acute advanced medical care and community/home care need to be considered [42]. Importantly, it is necessary to resolve transition-related issues, including those related to palliative care.

In this study, situation assessment and co-creation, identified as the nurses' roles in decision support, were new findings that should be considered in the SDM of critically ill patients. Situational assessment has mainly been used as a component of team activities aimed at patient safety on the healthcare frontline [43]. Moreover, in the process of clarifying the role of situation assessment, it was evident that nurses tend to assess the patient and the contextual situation from a "bird's-eye view" and that they play a collaborative role within the nursing team. This is because within the nursing community, nurses practice together and become

autonomous, which is a necessary process for them to build relationships with patients [44]. Nurses must play an autonomous role in conducting SDM in multidisciplinary settings.

The newly identified role of co-creation means that critical care nurses relate to the patient, family, and healthcare personnel and collaboratively create perspectives that cross various boundaries, as critical care nurses are involved in the patient's daily life after the chosen treatment has been implemented. Regardless of whether the prognosis is short or long term, the role of co-creating a desired life with the cooperation of those surrounding the patient is noteworthy. For example, a continuation of inotropic drugs during home transition does not reduce risk or improve long-term survival; however, it can improve the patient's health-related quality of life [45, 46]. Therefore, co-creation that is unbound by traditional treatment goals is necessary. The post-discharge life of critically ill patients who need substantial support is particularly susceptible to the social context. Recently, SDM has been viewed from the perspective of relational autonomy as opposed to individualistic, autonomous decision-making [47]. In the field of bioethics, a novel concept of relational autonomy has been proposed through various ethical approaches such as feminism and care ethics. It aims to resolve issues, such as those that require decisions to be made by individual patients and multiple parties, including family members and healthcare professionals. However, further consideration and development are needed to appropriately use and operationalize this concept [48, 49]. Nurses interact with vulnerable patients and distressed families in daily clinical practice. Hence, they are in a position to expand the concept of relational autonomy as practical knowledge [50]. The results of our study may contribute to further understanding of relational autonomy.

Our findings provide information to address issues that may occur when patients with severe heart failure participate in SDM and emphasize the role of nurses in SDM. For example, as presented in this study, critical care nurses can identify decision-making problems and offer decision-making support to address some of the difficulties faced by patients with severe heart failure during the process. Additionally, the quality of support for decision-making can be improved by utilizing decision-support roles within the nursing team. The continued clarification of the nuances of SDM and the decision-making process from a patient-family perspective is necessary, particularly within the microethnographic investigation of the actual decision-making process [39]. Furthermore, it is necessary to examine the concept of IP-SDM, wherein various healthcare professionals are involved in complex and difficult decision-making, while considering the scope of decision support and building evidence for this support based on clinical outcomes.

## Study limitations

There was a risk of self-selection bias, as participation in this study was voluntary, which made it easier for the CCNS interested in research topics to participate in the study. This study focused on critical care nurses who supported patients with severe heart failure in acute care hospitals in Japan. Hence, the study results may not be generalizable to the situations in other countries or non-critical care settings. However, many of the themes identified in this study revealed common issues reflected in the international literature on decision-making among patients with severe heart failure.

## Conclusions

This qualitative study found that patients with severe heart failure who needed to make critical care-related treatment decisions had difficulties that challenged their participation in SDM and the transition from advanced medical care. To improve IP-SDM, critical care nurses

should collaborate within the nursing team to provide decision-making support to patients by focusing on values and the search for meaning and to contribute to the healthcare team.

## Supporting information

**S1 Table. Additional quotes about patients' difficult situations in decision-making.** CCNS, certified nurse specialist(s) in critical care nursing; CRRT, continuous renal replacement therapy; ECMO, extracorporeal membrane oxygenation; HTx, heart transplantation; ICU, intensive care unit; VAD, ventricular assist device.
(DOCX)

**S2 Table. Additional quotes regarding the roles of critical care nurses in shared decision-making.** CCNS, certified nurse specialist(s) in critical care nursing; HTx, heart transplantation; ICU, intensive care unit; VAD, ventricular assist device.
(DOCX)

## Acknowledgments

We would like to thank Editage (www.editage.com) for English language editing.

## Author Contributions

**Conceptualization:** Noriko Inagaki.

**Data curation:** Noriko Inagaki.

**Formal analysis:** Noriko Inagaki, Natsuko Seto, Yuko Hayashi.

**Funding acquisition:** Noriko Inagaki.

**Investigation:** Noriko Inagaki.

**Project administration:** Noriko Inagaki.

**Supervision:** Natsuko Seto, Kumsun Lee, Yoshimitsu Takahashi, Takeo Nakayama, Yuko Hayashi.

**Visualization:** Noriko Inagaki.

**Writing – original draft:** Noriko Inagaki.

**Writing – review & editing:** Noriko Inagaki, Natsuko Seto, Kumsun Lee, Yoshimitsu Takahashi, Takeo Nakayama, Yuko Hayashi.

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
