## [Decision Letter · Decision Letter 0]

16 Apr 2023

PONE-D-23-05721The role of critical care nurses in shared decision-making for patients with severe heart failure: A qualitative studyPLOS ONE

Dear Dr. Inagaki,

Thank you for submitting your manuscript to PLOS ONE. After careful consideration, we feel that it has merit but does not fully meet PLOS ONE’s publication criteria as it currently stands. Therefore, we invite you to submit a revised version of the manuscript that addresses the points raised during the review process.

We look forward to receiving your revised manuscript.

Kind regards,

Dorothy Lall

Academic Editor

PLOS ONE

“I have read the journal's policy and the authors of this manuscript have the following competing interests: TN received research grants from I&H Co., Ltd., Cocokarafine Co., Ltd., and Konica Minolta Inc.; consulting fees from Otsuka Pharmaceutical; honoraria from Pfizer Japan INC., MSD K.K., Chugai Pharmaceutical Co., Takeda Pharmaceutical Co., Janssen Pharmaceutical K.K., Boehringer Ingelheim International GmbH, Eli Lilly Japan K.K., Maruho Co., Ltd., Mitsubishi Tanabe Pharma Co, Novartis Pharma K.K., Allergan Japan K.K., Maruho Co., Ltd., Novo Nordisk Pharma Ltd., TOA EIYO Ltd., Dentsu co., ONO PHARMACEUTICAL CO., LTD., GSK plc, Alexion Pharmaceuticals, Inc., and Cannon Medical Systems Co.; stock options from Bon Bon Inc.; donations from CancerScan and YUYAMA co. The other authors declare that they have no conflicts of interest to disclose.”

3. We noted in your submission details that a portion of your manuscript may have been presented or published elsewhere. [We noted in your submission details that a portion of your manuscript may have been presented or published elsewhere. [DETAILS AS NEEDED] Please clarify whether this [conference proceeding or publication] was peer-reviewed and formally published. If this work was previously peer-reviewed and published, in the cover letter please provide the reason that this work does not constitute dual publication and should be included in the current manuscript.] Please clarify whether this publication was peer-reviewed and formally published. If this work was previously peer-reviewed and published, in the cover letter please provide the reason that this work does not constitute dual publication and should be included in the current manuscript.

Additional Editor Comments:

well presented and an important inquiry. No specific suggestions to improve the manuscript.

Reviewers' comments:

Reviewer's Responses to Questions

**Comments to the Author**

1. Is the manuscript technically sound, and do the data support the conclusions?

Reviewer #1: Yes

2. Has the statistical analysis been performed appropriately and rigorously? 

Reviewer #1: N/A

3. Have the authors made all data underlying the findings in their manuscript fully available?

Reviewer #1: Yes

4. Is the manuscript presented in an intelligible fashion and written in standard English?

Reviewer #1: Yes

5. Review Comments to the Author

Reviewer #1: Dear authors,

thank you for opportunity to review this manuscript.

I believe the manuscript is well written and meets expected standards. The background and methods are well presented, results are clearly described, data support conclusions.

I only have minor comments for your consideration. The conclusions provided in abstract differ from the ones provided in manuscript. It might be clearer for readers if the abstract is revised.

Otherwise, I recommend article for publishing.

6. PLOS authors have the option to publish the peer review history of their article (what does this mean?). If published, this will include your full peer review and any attached files.

Reviewer #1: No

---

## [Author Response · Author response to Decision Letter 0]

21 May 2023

We deeply appreciate the constructive comment from reviewer #1. In response to your comment, we have further revised the manuscript for greater clarity. Our response is provided below.

Comment#1: The conclusions provided in abstract differ from the ones provided in manuscript. It might be clearer for readers if the abstract is revised.

Author’s response #1: 

Thank you for this suggestion. We have revised the conclusions in the abstract to ensure that they are consistent with the conclusion in the main text. We have also revised the aim in the abstract to accommodate the word limit.

Page 2, lines 42-43 and Page 3, lines 44-45 

Conclusions: Patients with severe heart failure have difficulty participating in shared decision-making. Critical care nurses should collaborate within the nursing team to improve interprofessional shared decision-making by providing decisional support to patients that focuses on values and meaning.

Page 2, lines 25-27

“This study aimed to clarify the perceptions of critical care nurses regarding situations with patients with severe heart failure that require difficult decision-making, and their role in supporting these patients.” 

Again, thank you very much for taking the time to review our paper.

---

## [Decision Letter · Decision Letter 1]

10 Jul 2023

The role of critical care nurses in shared decision-making for patients with severe heart failure: A qualitative study

PONE-D-23-05721R1

Dear Authors,

We’re pleased to inform you that your manuscript has been judged scientifically suitable for publication and will be formally accepted for publication once it meets all outstanding technical requirements.

Kind regards,

Dorothy Lall

Academic Editor

PLOS ONE

Additional Editor Comments (optional):

Reviewers' comments:

Reviewer's Responses to Questions

**Comments to the Author**

1. If the authors have adequately addressed your comments raised in a previous round of review and you feel that this manuscript is now acceptable for publication, you may indicate that here to bypass the “Comments to the Author” section, enter your conflict of interest statement in the “Confidential to Editor” section, and submit your "Accept" recommendation.

Reviewer #1: All comments have been addressed

2. Is the manuscript technically sound, and do the data support the conclusions?

Reviewer #1: Yes

3. Has the statistical analysis been performed appropriately and rigorously? 

Reviewer #1: N/A

4. Have the authors made all data underlying the findings in their manuscript fully available?

Reviewer #1: Yes

5. Is the manuscript presented in an intelligible fashion and written in standard English?

Reviewer #1: Yes

6. Review Comments to the Author

Reviewer #1: Dear authors,

I believe, you were able to address all comments.

I recommend article for publishing.

7. PLOS authors have the option to publish the peer review history of their article (what does this mean?). If published, this will include your full peer review and any attached files.

Reviewer #1: No

---

## [Editor Report · Acceptance letter]

12 Jul 2023

PONE-D-23-05721R1 

The role of critical care nurses in shared decision-making for patients with severe heart failure: A qualitative study 

Dear Dr. Inagaki:

I'm pleased to inform you that your manuscript has been deemed suitable for publication in PLOS ONE. Congratulations! Your manuscript is now with our production department. 

Kind regards, 

on behalf of

Dr. Dorothy Lall 

Academic Editor

PLOS ONE